# Interactive Structure Learning with Structural Query-by-Committee

**Christopher Tosh**
Columbia University
c.tosh@columbia.edu

**Sanjoy Dasgupta**
UC San Diego
dasgupta@cs.ucsd.edu

## Abstract

In this work, we introduce *interactive structure learning*, a framework that unifies many different interactive learning tasks. We present a generalization of the *query-by-committee* active learning algorithm for this setting, and we study its consistency and rate of convergence, both theoretically and empirically, with and without noise.

## 1   Introduction

We introduce *interactive structure learning*, an abstract problem that encompasses many interactive learning tasks that have traditionally been studied in isolation, including active learning of binary classifiers, interactive clustering, interactive embedding, and active learning of structured output predictors. These problems include variants of both supervised and unsupervised tasks, and allow many different types of feedback, from binary labels to must-link/cannot-link constraints to similarity assessments to structured outputs. Despite these surface differences, they conform to a common template that allows them to be fruitfully unified.

In interactive structure learning, there is a space of items $\mathcal{X}$—for instance, an input space on which a classifier is to be learned, or points to cluster, or points to embed in a metric space—and the goal is to learn a *structure* on $\mathcal{X}$, chosen from a family $\mathcal{G}$. This set $\mathcal{G}$ could consist, for example, of all linear classifiers on $\mathcal{X}$, or all hierarchical clusterings of $\mathcal{X}$, or all knowledge graphs on $\mathcal{X}$. There is a target structure $g^* \in \mathcal{G}$ and the hope is to get close to this target. This is achieved by combining a loss function or prior on $\mathcal{G}$ with interactive feedback from an expert.

We allow this interaction to be fairly general. In most interactive learning work, the dominant paradigm has been *question-answering*: the learner asks a question (like "what is the label of this point $x$?") and the expert provides the answer. We allow a more flexible protocol in which the learner provides a constant-sized *snapshot* of its current structure and asks whether it is correct ("does the clustering, restricted to these ten points, look right?"). If the snapshot is correct, the expert accepts it; otherwise, the expert fixes some part of it. This type of feedback, first studied in generality by [15], can be called *partial correction*. It is a strict generalization of question-answering, and as we explain in more detail below, it allows more intuitive interactions in many scenarios.

In Section 3, we present *structural query-by-committee*, a simple algorithm that can be used for any instance of interactive structure learning. It is a generalization of the well-known query-by-committee (QBC) algorithm [33, 16], and operates, roughly, by maintaining a posterior distribution over structures and soliciting feedback on snapshots on which there is high uncertainty. We also introduce an adaptation of the algorithm that allows convex loss functions to handle the noise. This helps computational complexity in some practical settings, most notably when $\mathcal{G}$ consists of linear functions, and also makes it possible to efficiently kernelize structural QBC.

In Section 4, we show that structural QBC is guaranteed to converge to the target $g^*$, even when the expert's feedback is noisy. In the appendix, we give rates of convergence in terms of a *shrinkage*

coefficient, present experiments on a variety of interactive learning tasks, and give an overview of related work.

## 2 Interactive structure learning

The space of possible interactive learning schemes is large and mostly unexplored. We can get a sense of its diversity from a few examples. In *active learning* [32], a machine is given a pool of unlabeled data and adaptively queries the labels of certain data points. By focusing on informative points, the machine may learn a good classifier using fewer labels than would be needed in a passive setting.

Sometimes, the labels are complex structured objects, such as parse trees for sentences or segmentations of images. In such cases, providing an entire label is time-consuming and it is easier if the machine simply suggests a label (such as a tree) and lets the expert either accept it or correct some particularly glaring fault in it. This is interaction with *partial correction*. It is more general than the *question-answering* usually assumed in active learning, and more convenient in many settings.

Interaction has also been used to augment *unsupervised* learning. Despite great improvements in algorithms for clustering, topic modeling, and so on, the outputs of these procedures are rarely perfectly aligned with the user's needs. The problem is one of underspecification: there are many legitimate ways to organize complex high-dimensional data, and no algorithm can magically guess which a user has in mind. However, a modest amount of interaction may help overcome this issue. For instance, the user can iteratively provide `must-link` and `cannot-link` constraints [37] to edit a flat clustering, or *triplet* constraints to edit a hierarchy [36].

These are just a few examples of interactive learning that have been investigated. The true scope of the settings in which interaction can be integrated is immense, ranging from structured output prediction to metric learning and beyond. In what follows, we aim to provide a unifying framework to address this profusion of learning problems.

### 2.1 The space of structures

Let $\mathcal{X}$ be a set of data points. This could be a pool of unlabeled data to be used for active learning, or a set of points to be clustered, or an entire instance space on which a metric will be learned.

We wish to learn a *structure* on $\mathcal{X}$, chosen from a class $\mathcal{G}$. This could, for instance, be the set of all labelings of $\mathcal{X}$ consistent with a function class $\mathcal{F}$ of classifiers (binary, multiclass, or with complex structured labels), or the set of all partitions of $\mathcal{X}$, or the set of all metrics on $\mathcal{X}$. Of these, there is some target $g^* \in \mathcal{G}$ that we wish to attain.

Although interaction will help choose a structure, it is unreasonable to expect that interaction alone could be an adequate basis for this choice. For instance, pinpointing a particular clustering over $n$ points requires $\Omega(n)$ must-link/cannot-link constraints, which is an excessive amount of interaction when $n$ is large.

To bridge this gap, we need a prior or a loss function over structures. For instance, if $\mathcal{G}$ consists of flat $k$-clusterings, then we may prefer clusterings with low $k$-means cost. If $\mathcal{G}$ consists of linear separators, then we may prefer functions with small norm $\|g\|$. In the absence of interaction, the machine would simply pick the structure that optimizes the prior or cost function. In this paper, we assume that this preference is encoded as a prior distribution $\pi$ over $\mathcal{G}$.

We emphasize that although we have adopted a Bayesian formulation, there is no assumption that the target structure $g^*$ is actually drawn from the prior.

### 2.2 Feedback

We consider schemes in which each individual round of interaction is not expected to take too long. This means, for instance, that the expert cannot be shown an entire clustering, of unrestricted size, and asked to comment upon it. Instead, he or she can only be given a small *snapshot* of the clustering, such as its restriction to 10 elements. The feedback on this snapshot will be either be to accept it, or to provide some constraint that fixes part of it.

In order for this approach to work, it is essential that structures be *locally checkable*: that is, $g$ corresponds to the target $g^*$ if and only if every snapshot of $g$ is satisfactory.

When $g$ is a clustering, for instance, the snapshots could be restrictions of $g$ to subsets $S \subseteq \mathcal{X}$ of some fixed size $s$. Technically, it is enough to take $s = 2$, which corresponds to asking the user questions of the form 'Do you agree with having `zebra` and `giraffe` in the same cluster?" From the viewpoint of human-computer interaction, it might be preferable to use larger subsets (like $s = 5$ or $s = 10$), with questions such as "Do you agree with the clustering $\{$`zebra`, `giraffe`, `dolphin`$\}$, $\{$`whale`, `seal`$\}$?" Larger substructures provide more context and are more likely to contain glaring faults that the user can easily fix (`dolphin` and `whale` must go together). In general, we can only expect the user to provide partial feedback in these cases, rather than fully correcting the substructure.

## 2.3 Snapshots

Perhaps the simplest type of snapshot of a structure $g$ is *the restriction of $g$ to a small number of points*. We start by discussing this case, and later present a generalization.

### 2.3.1 Projections

For any $g \in \mathcal{G}$ and any subset $S \subseteq \mathcal{X}$ of size $s = O(1)$, let $g|_S$ be a suitable notion of the restriction of $g$ to $S$, which we will sometimes call the *projection* of $g$ onto $S$. For instance:

- $\mathcal{G}$ is a set of classifiers on $\mathcal{X}$. We can take $s = 1$ and let $g|_x$ be $(x, g(x))$ for any $x \in \mathcal{X}$.
- $\mathcal{G}$ is a set of partitions (flat clusterings) of $\mathcal{X}$. For a set $S \subseteq \mathcal{X}$ of size $s \geq 2$, let $g|_S$ be the induced partition on just the points $S$.

As discussed earlier, it will often be helpful to pick projections of size larger than the minimal possible $s$. For clusterings, for instance, any $s \geq 2$ satisfies local checkability, but human feedback might be more effective when $s = 10$ than when $s = 2$. Thus, in general, the queries made to the expert will consist of snapshots (projections of size $s = 10$, say) that can in turn be decomposed further into *atomic units* (projections of size 2).

### 2.3.2 Atomic decompositions of structures

Now we generalize the notion of projection to other types of snapshots and their atomic units.

We will take a functional view of the space of structures $\mathcal{G}$, in which each structure $g$ is specified by its "answers" to a set of *atomic questions* $\mathcal{A}$. For instance, if $\mathcal{G}$ is the set of partitions of $\mathcal{X}$, then $\mathcal{A} = \binom{\mathcal{X}}{2}$, with $g(\{x, x'\}) = 1$ if $g$ places $x, x'$ in the same cluster and 0 otherwise.

The queries made during interaction can, in general, be composed of multiple atomic units, and feedback will be received on at least one of these atoms. Formally, let $\mathcal{Q}$ be the space of queries. In the partitioning example, this might be $\binom{\mathcal{X}}{10}$. The relationship between $\mathcal{Q}$ and $\mathcal{A}$ is captured by the following requirements:

- Each $q \in \mathcal{Q}$ can be decomposed as a set of atomic questions $A(q) \subseteq \mathcal{A}$, and we write $g(q) = \{(a, g(a)) : a \in A(q)\}$. In the partitioning example, $A(q)$ is the set of all pairs in $q$.
- The user accepts $g(q)$ if and only if $g$ satisfactorily answers every atomic question in $q$, that is, if and only if $g(a) = g^*(a)$ for all $a \in A(q)$.

## 2.4 Summary of framework

To summarize, interactive structure learning has two key components:

- *A reduction to multiclass classifier learning.* We view each structure $g \in \mathcal{G}$ as a function on atomic questions $\mathcal{A}$. Thus, learning a good structure is equivalent to picking one whose labels $g(a)$ are correct.
- *Feedback by partial correction.* For practical reasons we consider broad queries, from a set $\mathcal{Q}$, where each query can be decomposed into atomic questions, allowing for partial corrections. This decomposition is given by the function $A : \mathcal{Q} \to 2^{\mathcal{A}}$.

---

**Algorithm 1** STRUCTURAL QBC

---

**Input:** Distribution[1] $\nu$ over query space $\mathcal{Q}$ and initial prior distribution $\pi_o$ over $\mathcal{G}$
**Output:** Posterior distribution $\pi_t$ over $\mathcal{G}$
**for** $t = 1, 2, \ldots$ **do**
    Draw $g_t \sim \pi_{t-1}$
    **while** Next query $q_t$ has not been chosen **do**
        Draw $q \sim \nu$ and $g, g' \sim \pi_{t-1}$
        With probability $d(g, g'; q)$: take $q_t = q$
    **end while**
    Show user $q_t$ and $g_t(q_t)$ and receive feedback in form of pairs $(a_t, y_t)$
    Update posterior: $\pi_t(g) \propto \pi_{t-1}(g) \exp(-\beta \cdot \mathbf{1}(g(a_t) \neq y_t))$
**end for**

---

The reduction to multiclass classification immediately suggests algorithms that can be used in the interactive setting. We are particular interested in *adaptive* querying, with the aim of finding a good structure with minimal interaction. Of the many schemes available for binary classifiers, one that appears to work well in practice and has good statistical properties is *query-by-committee* [33, 16]. It is thus a natural candidate to generalize to the broader problem of structure learning.

## 3 Structural QBC

Query-by-committee, as originally analyzed by [16], is an active learning algorithm for binary classification in the noiseless setting. It uses a prior probability distribution $\pi$ over its classifiers and keeps track of the current version space, i.e. the classifiers consistent with the labeled data seen so far. At any given time, the next query is chosen as follows:

- Repeat:
  - Pick $x \in \mathcal{X}$ at random
  - Pick classifiers $h, h'$ at random from $\pi$ restricted to the current version space
  - If $h(x) \neq h'(x)$: halt with $x$ as the query

In our setting, the feedback at time $t$ is the answer $y_t$ to some atomic question $a_t \in \mathcal{A}$, and we can define the resulting version space to be $\{g \in \mathcal{G} : g(a_{t'}) = y_{t'} \text{ for all } t' \leq t\}$. The immediate generalization of QBC would involve picking a query $q \in \mathcal{Q}$ at random (or more generally, drawn from some query distribution $\nu$), and then choosing it if $g, g'$ sampled from $\pi$ restricted to our version space happen to disagree on it. But this is unlikely to work well, because the answers to queries are no longer binary labels but mini-structures. As a result, $g, g'$ are likely to disagree on minor details even when the version space is quite small, leading to excessive querying. To address this, we will use a more refined notion of the difference between $g(q)$ and $g'(q)$:

$$d(g, g'; q) = \frac{1}{|A(q)|} \sum_{a \in A(q)} \mathbf{1}[g(a) \neq g'(a)].$$

In words, this is the fraction of atomic subquestions of $q$ on which $g$ and $g'$ disagree. It is a value between 0 and 1, where higher values mean that $g(q)$ differs significantly from $g'(q)$. Then we will query $q$ with probability $d(g, g'; q)$.

### 3.1 Accommodating noisy feedback

We are interested in the noisy setting, where the user's feedback may occasionally be inconsistent with the target structure. In this case, the notion of a version space is less clear-cut. Our modification is very simple: the feedback at time $t$, say $(a_t, y_t)$, causes the posterior to be updated as follows:

$$\pi_t(g) \propto \pi_{t-1}(g) \exp(-\beta \cdot \mathbf{1}[g(a_t) \neq y_t]). \tag{1}$$

Here $\beta > 0$ is a constant that controls how aggressively errors are punished. In the noiseless setting, we can take $\beta = \infty$ and recover the original QBC update. Even with noise, however, this posterior update still enjoys nice theoretical properties. The full algorithm is shown in Algorithm 1.

## 3.2 Uncertainty and informative queries

What kinds of queries will structural QBC make? To answer this, we first quantify the *uncertainty* in the current posterior about a particular query or atom. Define the uncertainty of atom $a \in \mathcal{A}$ under distribution distribution $\pi$ as $u(a; \pi) = \Pr_{g, g' \sim \pi}(g(a) \neq g'(a))$ and $u(q; \pi)$ as the average uncertainty of its atoms $A(q)$. These values lie in the range $[0, 1]$.

The probability that a particular query $q \in \mathcal{Q}$ is chosen in round $t$ by structural QBC is proportional to $\nu(q)u(q; \pi_{t-1})$. Thus, queries with higher uncertainty under the current posterior are more likely to be chosen. As the following lemma demonstrates, getting feedback on uncertain atoms eliminates, or down-weights in the case of noisy feedback, many structures inconsistent with $g^*$.

**Lemma 1.** *For any distribution $\pi$ over $\mathcal{G}$, we have $\pi(\{g : g(a) \neq y\}) \geq u(a; \pi)/2$.*

The proof of Lemma 1 is deferred to the appendix. This gives some intuition for the query selection criterion of structural QBC, and will later be used in the proof of consistency.

## 3.3 General loss functions

The update rule for structural QBC, equation (1), results in a posterior of the form $\pi_t(g) \propto \pi(g) \exp(-\beta \cdot \#(\text{mistakes made by } g))$, which may be difficult to sample from. Thus, we consider a broader class of updates,

$$\pi_t(g) \propto \pi_{t-1}(g) \exp(-\beta \cdot \ell(g(a_t), y_t)), \tag{2}$$

where $\ell(\cdot, \cdot)$ is a general loss function. In the special case where $\mathcal{G}$ consists of linear functions and $\ell$ is convex, $\pi_t$ will be a log-concave distribution, which allows for efficient sampling [28]. We will show that this update also enjoys nice theoretical properties, albeit under different noise conditions.

To formally specify this setting, let $\mathcal{Y}$ be the space of answers to atomic questions $\mathcal{A}$, and suppose that structures in $\mathcal{G}$ generate values in some prediction space $\mathcal{Z} \subseteq \mathbb{R}^d$. That is, each $g \in \mathcal{G}$ is a function $g : \mathcal{A} \to \mathcal{Z}$, and any output $z \in \mathcal{Z}$ gets translated to some prediction in $\mathcal{Y}$. The loss associated with predicting $z$ when the true answer is $y$ is denoted $\ell(z, y)$. Here are some examples:

- $0 - 1$ loss. $\mathcal{Z} = \mathcal{Y}$ and $\ell(z, y) = \mathbf{1}(y \neq z)$.
- Squared loss. $\mathcal{Y} = \{-1, 1\}$, $\mathcal{Z} = [-B, B]$, and $\ell(z, y) = (y - z)^2$.
- Logistic loss. $\mathcal{Y} = \{-1, 1\}$, $\mathcal{Z} = [-B, B]$ for some $B > 0$, and $\ell(z, y) = \ln(1 + e^{-yz})$.

When moving from a discrete to a continuous prediction space, it becomes very possible that the predictions, on a particular atom, of two randomly chosen structures will be close but not perfectly aligned. Thus, instead of checking strict equality of these predictions, we need to modify our querying strategy to take into account the distance between them. To this end, we will use the normalized average squared Euclidean distance:

$$d^2(g, g'; q) = \frac{1}{|A(q)|} \sum_{a \in A(q)} \frac{\|g(a) - g'(a)\|^2}{D}$$

where $D = \max_{a \in \mathcal{A}} \max_{g, g' \in \mathcal{G}} \|g(a) - g'(a)\|^2$. Note that $d^2(g, g'; q)$ is a value between 0 and 1. We treat it as a probability, in exactly the same way we used $d(g, g'; q)$ in the 0-1 loss setting.

In the 0-1 loss setting, structural QBC chooses queries proportional to their uncertainty. What queries will structural QBC make in the general loss setting? Define the variance of $a \in \mathcal{A}$ under $\pi$ as

$$\text{var}(a; \pi) = \frac{1}{2} \sum_{g, g' \in \mathcal{G}} \pi(g) \pi(g') \|g(a) - g'(a)\|^2$$

and $\text{var}(q; \pi)$ as the average variance of its atoms $A(q)$. Then the probability that structural QBC chooses $q \in \mathcal{Q}$ at step $t$ is proportional to $\nu(q)\text{var}(q; \pi_{t-1})$ in the general loss setting.

**Algorithm 2** ROBUST QUERY SELECTION

---

**Input:** Fixed set of queries $q_1, \ldots, q_m \in \mathcal{Q}$, current distribution $\pi$ over $\mathcal{G}$
**Output:** Query $q_i$
Initial shrinkage estimate: $\widehat{u}_o = 1/2$
**for** $t = 0, 1, 2, \ldots$ **do**
    Draw $g_1, g_1', \ldots, g_{n_t}, g_{n_t}' \sim \pi$
    If there exists $q_j$ such that $\frac{1}{n_t} \sum_{i=1}^{n_t} d(g_i, g_i'; q_j) \geq \widehat{u}_t$ then we halt and query $q_j$
    Otherwise, let $\widehat{u}_{t+1} = \widehat{u}_t/2$.
**end for**

---

## 3.4 Kernelizing structural QBC

Consider the special case where $\mathcal{G}$ consists of linear functions, i.e. $\mathcal{G} = \{g_w(x) = \langle x, w \rangle : w \in \mathbb{R}^d\}$. As mentioned above, when our loss function is convex, the posteriors we encounter are log-concave, and thus efficiently samplable. But what if we want a more expressive class than linear functions? To address this, we will resort to kernels.

Gilad-Bachrach et al. [17] investigated the use of kernels in QBC. In particular, they observed that QBC does not actually need samples from the prior restricted to the current version space. Rather, given a candidate query $x$, it is enough to be able to sample from the distribution the posterior induces over the labelings of $x$. Although their work was in the realizable binary setting, this observation still applies to our setting.

Given a feature mapping $\phi : \mathcal{X} \rightarrow \mathbb{R}^d$, our posterior update becomes $\pi_t(g_w) \propto \pi_{t-1}(g_w) \exp\left(-\beta \ell(\langle \phi(x_t), w \rangle, y_t)\right)$. As the following lemma shows, when $\ell(\cdot, \cdot)$ is the squared-loss and our prior is Gaussian, the predictions of the posterior have a univariate normal distribution.

**Lemma 2.** *Suppose* $\pi = \mathcal{N}(0, \sigma_o^2 I_d)$, $\ell(\cdot, \cdot)$ *is the squared-loss, and we have observed* $(x_1, y_1), \cdots, (x_t, y_t)$. *If* $g_w \sim \pi_t$, *then* $\langle w, \phi(x) \rangle \sim \mathcal{N}(\mu, \sigma^2)$ *where*

$$\mu = 2\sigma_o^2 \beta \kappa^T \left(I_t - \Sigma_o K\right) y \qquad and \qquad \sigma^2 = \sigma_o^2 \left(\phi(x)^T \phi(x) - \kappa^T \Sigma_o \kappa\right)$$

*for* $K_{ij} = \langle \phi(x_i), \phi(x_j) \rangle$, $\kappa_i = \langle \phi(x_i), \phi(x) \rangle$, *and* $\Sigma_o = \left(\frac{1}{2\sigma^2 \beta} I_t + K\right)^{-1}$.

The proof is deferred to the appendix. The important observation here is that all the quantities involving the feature mapping in Lemma 2 are inner products. Thus we never need to explicitly construct any feature vectors.

## 3.5 Reducing the randomness in structural QBC

It is easy to see that the query selection procedure of structural QBC is a rejection sampler where each query $q$ is chosen with probability proportional to $\nu(q)u(q; \pi_t)$ (in the case of the 0-1 loss) or $\nu(q)\mathrm{var}(q; \pi_t)$ (for general losses). However, it is possible for the rejection rate to be quite high, even when there are many queries that have much higher uncertainty or variance than the rest. To circumvent this issue, we introduce a 'robust' version of structural QBC, wherein many candidate queries are sampled, and the query that has the highest uncertainty or variance is chosen.

In the 0-1 loss case, we can estimate the uncertainty of a candidate query $q$ by drawing many pairs $g_1, g_1', \ldots, g_n, g_n' \sim \pi_t$ and using the unbiased estimator $\widehat{u}(q; \pi_t) := \frac{1}{n} \sum_{i=1}^{n} d(g_i, g_i'; q)$.

Unfortunately, the number of structures we need to sample in order to identify the most uncertain query depends on its uncertainty, which we do not know a priori. To circumvent this difficulty, we can use the halving procedure shown in Algorithm 2. If the appropriate number of structures are sampled at each round $t$, on the order of $O((1/u_t^2) \log(m \log(1/u_o)))$ for some crude lower bound $u_o$ on the highest uncertainty, then with high probability this procedure terminates with a candidate query whose uncertainty is within a constant factor of the highest uncertainty [35].

# 4 Consistency of structural QBC

In this section, we look at a typical setting in which there is a finite but possibly very large pool of candidate questions $\mathcal{Q}$, and thus the space of structures $\mathcal{G}$ is effectively finite. Let $g^* \in \mathcal{G}$ be the target structure, as before. Our goal in this setting is to demonstrate the *consistency* of structural QBC, meaning that $\lim_{t \to \infty} \pi_t(g^*) = 1$ almost surely. To do so, we formalize our setting. Note that the random outcomes during time step $t$ of structural QBC consist of the query $q_t$, the atomic question $a_t$ that the expert chooses to answer (pick one at random if the expert answers several of them), and the response $y_t$ to $a_t$. Let $\mathcal{F}_t$ denote the sigma-field of all outcomes up to, and including, time $t$.

## 4.1 Consistency under 0-1 loss

In order to prove consistency, we will have to make some assumptions about the feedback we receive from a user. For query $q \in \mathcal{Q}$ and atomic question $a \in A(q)$, let $\eta(y|a, q)$ denote the conditional probability that the user answers $y$ to atomic question $a$, in the context of query $q$. Our first assumption is that the single most likely answer is $g^*(a)$.

**Assumption 1.** *There exists $0 < \lambda \leq 1$ such that $\eta(g^*(a)|a, q) - \eta(y|a, q) \geq \lambda$ for all $q \in \mathcal{Q}$ and $a \in A(q)$ and all $y \neq g^*(a)$.*

(We will use the convention $\lambda = 1$ for the noiseless setting.) In the learning literature, Assumption 1 is known as Massart's bounded noise condition [2].

The following lemma, whose proof is deferred to the appendix, demonstrates that under Assumption 1, the posterior probability of $g^*$ increases in expectation with each query, as long as the $\beta$ parameter of the update rule in equation (1) is small enough relative to $\lambda$.

**Lemma 3.** *Fix any $t$ and suppose the expert provides an answer to atomic question $a_t \in A(q_t)$ at time $t$. Let $\gamma_t = \pi_{t-1}(\{g \in \mathcal{G} : g(a_t) = g^*(a_t)\})$. Define $\Delta_t$ by:*

$$\mathbb{E}\left[\frac{1}{\pi_t(g^*)} \,\middle|\, \mathcal{F}_{t-1}, q_t, a_t\right] = (1 - \Delta_t)\frac{1}{\pi_{t-1}(g^*)},$$

*Under Assumption 1, $\Delta_t$ can be lower-bounded as follows:*

*(a) If $\lambda = 1$ (noiseless setting), $\Delta_t \geq (1 - \gamma_t)(1 - e^{-\beta})$.*

*(b) For $\lambda \in (0, 1)$, if $\beta \leq \lambda/2$, then $\Delta_t \geq \beta\lambda(1 - \gamma_t)/2$.*

To understand the requirement $\beta = O(\lambda)$, consider an atomic question on which there are just two possible labels, 1 and 2, and the expert chooses these with probabilities $p_1$ and $p_2$, respectively. If the correct answer according to $g^*$ is 1, then $p_1 \geq p_2 + \lambda$ under Assumption 1. Let $\mathcal{G}_2$ denote structures that answer 2.

- With probability $p_1$, the expert answers 1, and the posterior mass of $\mathcal{G}_2$ is effectively multiplied by $e^{-\beta}$.

- With probability $p_2$, the expert answers 2, and the posterior mass of $\mathcal{G}_2$ is effectively multiplied by $e^{\beta}$.

The second outcome is clearly undesirable. In order for it to be counteracted, in expectation, by the first, $\beta$ must be small relative to $p_1/p_2$. The condition $\beta \leq \lambda/2$ ensures this.

Lemma 3 does not, in itself, imply consistency. It is quite possible for $1/\pi_t(g^*)$ to keep shrinking but not converge to 1. Imagine, for instance, that the input space has two parts to it, and we keep improving on one of them but not the other. What we need is, first, to ensure that the queries $q_t$ capture some portion of the uncertainty in the current posterior, and second, that the user chooses atoms that are at least slightly informative. The first condition is assured by the SQBC querying strategy. For the second, we need an assumption.

**Assumption 2.** *There is some minimum probability $p_o > 0$ for which the following holds. If the user is presented with a query $q$ and a structure $g \in \mathcal{G}$ such that $g(q) \neq g^*(q)$, then with probability at least $p_o$ the user will provide feedback on some $a \in A(q)$ such that $g(a) \neq g^*(a)$.*

Assumption 2 is one way of avoiding scenarios in which a user never provides feedback on a particular atom $a$. In such a pathological case, we might not be able to recover $g^*(a)$, and thus our posterior will always put some probability mass on structures that disagree with $g^*$ on $a$.

The following lemma gives lower bounds on $1 - \gamma_t$ under Assumption 2.

**Lemma 4.** *Suppose that $\mathcal{G}$ is finite and the user's feedback obeys Assumption 2. Then there exists a constant $c > 0$ such that for every round $t$*

$$\mathbb{E}[1 - \gamma_t \mid \mathcal{F}_{t-1}] \geq c\,\pi_{t-1}(g^*)^2(1 - \pi_{t-1}(g^*))^2$$

*where $\gamma_t = \pi_{t-1}(\{g \in \mathcal{G} : g(a_t) = g^*(a_t)\})$ and $a_t$ is the atom the user provides feedback on.*

Together, Lemmas 3 and 4 show that the sequence $\frac{1}{\pi_t(g^*)}$ is a positive supermartingale that decreases in expectation at each round by an amount that depends on $\pi_t(g^*)$. The following lemma tells us exactly when such stochastic processes can be guaranteed to converge.

**Lemma 5.** *Let $f : [0,1] \to \mathbb{R}_{\geq 0}$ be a continuous function such that $f(1) = 0$ and $f(x) > 0$ for all $x \in (0,1)$. If*

$$\mathbb{E}\left[\frac{1}{\pi_t(g^*)} \,\middle|\, \mathcal{F}_{t-1}\right] \leq \frac{1}{\pi_{t-1}(g^*)} - f(\pi_{t-1}(g^*))$$

*for each $t \in \mathbb{N}$, then $\pi_t(g^*) \to 1$ almost surely.*

As a corollary, we see that structural QBC is consistent.

**Theorem 6.** *Suppose that $\mathcal{G}$ is finite, and Assumptions 1 and 2 hold. If $\pi(g^*) > 0$, then $\pi_t(g^*) \to 1$ almost surely under structural QBC's query strategy.*

We provide a proof of Theorem 6 in the appendix, where we also provide rates of convergence.

## 4.2 Consistency under general losses

We now turn to analyzing structural QBC with general losses. As before, we will need to make some assumptions. The first is that the loss function is well-behaved.

**Assumption 3.** *The loss function is bounded, $0 \leq \ell(z,y) \leq B$, and Lipschitz in its first argument, i.e. $\ell(z,y) - \ell(z',y) \leq C\|z - z'\|$, for some constants $B, C > 0$.*

It is easily checked that this assumption holds for the three loss functions we mentioned earlier.

In the case of 0-1 loss, we assumed that for any atomic question $a$, the correct answer $g^*(a)$ would be given with higher probability than any incorrect answer. We now formulate an analogous assumption for the case of more general loss functions. Recall that $\eta(\cdot|a)$ is the conditional probability distribution over the user's answers to $a \in \mathcal{A}$ (we can also allow $\eta$ to also depend upon the context $q$, as we did before; here we drop the dependence for notational convenience). The expected loss incurred by $z \in \mathcal{Z}$ on this atom is thus

$$L(z,a) = \sum_y \eta(y|a)\,\ell(z,y).$$

We will require that for any atomic question $a$, this expected loss is minimized when $z = g^*(a)$, and predicting any other $z$ results in expected loss that grows with the distance between $z$ and $g^*(a)$.

**Assumption 4.** *There exists a constant $\lambda > 0$ such that $L(z,a) - L(g^*(a), a) \geq \lambda\|z - g^*(a)\|^2$ for any atomic question $a \in \mathcal{A}$ and any $z \in \mathcal{Z}$.*

Let's look at some concrete settings:

- $0-1$ loss with $\mathcal{Y} = \mathcal{Z} = \{0,1\}$. Assumption 4 is equivalent to Assumption 1.

- Squared loss with $\mathcal{Y} = \{-1,1\}$ and $\mathcal{Z} \subset \mathbb{R}$. Assumption 4 is satisfied when $g^*(a) = \mathbb{E}[y|a]$ and $\lambda = 1$.

- Logistic loss with $\mathcal{Y} = \{-1,1\}$ and $\mathcal{Z} = [-B, B]$. For $a \in A$, let $p = \eta(1|a)$. Assumption 4 is satisfied when $g^*(a) = \ln\frac{p}{1-p}$ and $\lambda = \frac{2e^{2B}}{(1+e^B)^4}$.

From these examples, it is clear that requiring $g^*(a)$ to be the minimizer of $L(z, a)$ is plausible if $\mathcal{Z}$ is a discrete space but much less so if $\mathcal{Z}$ is continuous. In general, we can only hope that this holds approximately. With this caveat in mind, we stick with Assumption 4 as a useful but idealized mathematical abstraction.

With these assumptions in place, the following theorem guarantees the consistency of structural QBC under general losses. Its proof is deferred to the appendix.

**Theorem 7.** *Suppose we are in the general loss setting, $\mathcal{G}$ is finite, and the user's feedback satisfies Assumptions 2, 3, and 4. If $\pi(g^*) > 0$, then $\pi_t(g^*) \to 1$ almost surely.*

## 5   Conclusion

In this work, we introduced interactive structure learning, a generic framework for learning structures under partial correction feedback. This framework can be applied to any structure learning problem in which structures are in one-to-one correspondence with their answers to atomic questions. Thus, interactive structure learning may be viewed as a generalization of active learning, interactive clustering with pairwise constraints, interactive hierarchical clustering with triplet constraints, and interactive ordinal embeddings with quadruplet constraints.

On the algorithmic side, we introduced structural QBC, a generalization of the classical QBC algorithm to the interactive structure learning setting. We demonstrated that this algorithm is consistent, even in the presence of noise, provided that we can sample from a certain natural posterior. In the appendix, we also provided rates of convergence. Because this posterior is often intractable to sample from, we also considered an alternative posterior based on convex loss functions that sometimes allows for efficient sampling. We showed that structural QBC remains consistent in this setting, albeit under different noise conditions.

In the appendix, we provide experiments on both interactive clustering and active learning tasks. On the interactive clustering side, these experiments demonstrate that even when the prior distribution places relatively low mass on the target clustering, structural QBC is capable of recovering a low-error clustering with relatively few rounds of interaction. In contrast, these experiments also show that random corrections are not quite as useful. On the active learning side, there are experiments demonstrating the good empirical performance of structural QBC using linear classifiers with the squared-loss posterior update, with and without kernelization.

### Acknowledgments

The authors are grateful to the reviewers for their feedback and to the NSF for support under grant CCF-1813160. Part of this work was done at the Simons Institute for Theoretical Computer Science, Berkeley, during the "Foundations of Machine Learning" program. CT also thanks Stefanos Poulis and Sharad Vikram for helpful discussions and feedback.

## Footnotes

[1]In the setting where $\mathcal{Q}$ is finite, a reasonable choice of $\nu$ would be uniform over $\mathcal{Q}$.

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
