[Reviews · NeurIPS 2018]

Reviewer 1



This paper formulates a framework which unifies several interactive learning problems with a structure such as interactive clustering. Next, the authors show that QBC can be generalized and kernelized to solve the problems in the framework. The consistency and rate of convergence are analyzed. I do not think I could judge the novelty of the theorems and proofs well. Thus, my comments focus on the practicality of the proposed algorithm, which I believe is relevant to its significance and the impact to the field. Strength: I think the direction of applying QBC to interactive learning settings other than active learning is promising. QBC might not be that practical in active learning problems because state-of-the-art models are usually neural network and modern neural networks have huge version space and there are lots of very different and equally good local minimums. It is also not easy to sample from neural networks efficiently (dropout only define the uncertainty on a small number of weights). However, the state-of-the-art clustering methods like GMM are usually very simple and easy to sample. The writing is also very clear. Reading the paper is like reading a chapter of a textbook. Weakness (or more like suggestions for future improvements): Although paper tries to address the practical concerns (handling the noisy cases and only need to sample two parameters at a time), the method still has some practical issues. For example, QBC-based methods are vulnerable to noise. If we could quantify the level of noise at the beginning (e.g., knows the beta in eq. (1)), we can modify QBC to have nice guarantees. However, we usually do not know this information before collecting the data. In fact, most of the time, we do not even know which class of models with what kinds of hyper-parameters will perform very well before collecting enough data. Furthermore, clustering tasks are usually more ambiguous. There might also be many equally good local minimums in complicated clustering problems and models. Different annotators often have their own solutions from different local minimums. This means the noise level is usually high and hard to be predicted. Without knowing beta, I do not know how robust the structure QBC would be. Even if we know the noise level, it is possible that we spend lots of effort searching g* by eliminating lots of equally good local minimums, but we find that g* is not good enough eventually because the model assumption has some flaws or the original model becomes too simple as more samples are collected. The experiments are only done using toy data and do not compare with other strategies such as uncertainty sampling. Is the algorithm scalable? Is it possible to know the time complexity of the proposed structural QBC when choosing the next example with respect to all the hyper-parameters? Can this study really motivate more people to use QBC in practice? I understand that this is a theoretical paper, and my educated guess is that the theoretical part of the paper is very solid even though the practicality of the method has not been fully demonstrated. That's the main reason why I vote a clear accept. Minor suggestions: 1. I can only find the explanation of notation \nu in Algorithm 1. Suggest mentioning the meaning of \nu in the text as well. 2. In Algorithm 2, it does not say how to determine n_t. What does the "appropriate number" mean in line 225? It is hard to find the answer in [30]. 3. Line 201, it should be Gilad-Bachrach et al. [16] investigated ... 4. In the related work section, I recommend the authors to cite some other efforts, especially the ones which try to make QBC based methods more practical and test its performances on multiple real-world datasets (e.g., Huang et al.) Huang, Tzu-Kuo, et al. "Efficient and parsimonious agnostic active learning." Advances in Neural Information Processing Systems. 2015. After rebuttal I think all reviewers agree that this is a solid theoretical work, so my judgment remains the same. The authors do not address the practicality issues in the rebuttal which prevents me to increase the overall score to an even higher level.

Reviewer 2



This paper proposes and develops a novel framework for interactive structure learning, and in particular presents a generalization of the "query by committee" method for active learning for this new setting. The paper then presents fundamental theoretical results such as consistency and rate of convergence for that generalized method. The paper addresses an important problem, namely that of interactive learning of structures, which generalizes the typical active learning framework in a number of ways, and the formulation they develop is well motivated. Judging from the detailed proofs in the supplemental materials, the theoretical analyses are sound. The supplemental materials also contain considerable empirical results, which adds to the audience's confidence in the general effectiveness of the proposed approach. It would be nice, therefore, if they could include some brief summary of the experiment section in the paper itself, if at all possible. Overall a good paper with significant contributions on a general problem of practical import.

Reviewer 3



Main idea The paper introduces a new framework of interactive structure learning that focuses on interactive learning on partial correction. The framework formulate interactive queries as a bag of atomic questions, and user correction corresponds to picking one atomic question with an error. Seeing it as an extension to multilabel classification, the paper generalizes the query-by-committee algorithm to the framework by using priors and weighted updates. The paper analyzes QBC both under perfect and noisy feedbacks, and under 0-1 and general convex losses. The authors prove the convergence of specified algorithm in main text, while experiments and rates of convergence is given in the appendix. Comments The paper presents a nice framework that summarizes both multi-label classification and hierarchical clustering. I feel it is good fit for publication at NIPS. Several comments and directions: 1. The first line of Algorithm 1 mentions a prior distribution over query space, which is not formally defined in main text (the prior on structures is defined). I feel it benefitial to formally define it and discuss the impact of it on convergence (rate). 2. There is a line of supervised interactive learning that can be mentioned in the related works section. Theses works uses additional information as feedback and is more related to the current work than pure active learning. E.g., Poulis, S. and Dasgupta, S. Learning with feature feedback: From theory to practice. Xu, Y., Zhang, H., Miller, K., Singh, A., and Dubrawski, A. Noise-tolerant interactive learning from pairwise compar- isons with near-minimal label complexity 3. A conclusion is probably needed for final version. 4. The paper considers Massart noise for noisy feedback. It would be interesting to consider other choices, like Tsybakov noise or adversarial noise. In this situation convergence might not be possible, but still some other conclusions might be drawn. Quality The paper is well written and the analysis seems complete. Clarity The paper is mostly clear. It does not have conclusion part - I feel some polishing is needed for final version. Originality The paper is original. Significance The paper gives an important framework that combines multi-label classification and clustering. Update: I've read the authors' response and I feel satisfied about it.